# Prevalence, Serotypes, and Antimicrobial Resistance Patterns of Non-Typhoid *Salmonella* in Food in Northern Taiwan

**DOI:** 10.3390/pathogens11060705

**Published:** 2022-06-18

**Authors:** Yi-Jung Chang, Chyi-Liang Chen, Hsin-Ping Yang, Cheng-Hsun Chiu

**Affiliations:** 1Division of Pediatric Infectious Diseases, Department of Pediatrics, Chang Gung Memorial Hospital, Chang Gung University College of Medicine, Taoyuan 33302, Taiwan; r64321@cgmh.org.tw; 2Molecular Infectious Disease Research Center, Chang Gung Memorial Hospital, Taoyuan 33305, Taiwan; dinoschen@adm.cgmh.org.tw (C.-L.C.); yhp0903@cgmh.org.tw (H.-P.Y.); 3Department of Microbiology and Immunology, School of Medicine, College of Medicine, Chang Gung University, Taoyuan 33302, Taiwan; 4Graduate Institute of Clinical Medical Sciences, Chang Gung University College of Medicine, Taoyuan 33302, Taiwan

**Keywords:** *Salmonella*, antimicrobial resistance, multi-locus sequence typing, retail pork, chicken, multidrug-resistant

## Abstract

*Salmonella* is one of the most common bacteria causing food poisoning worldwide. We evaluated the prevalence, the serotypes, and the antimicrobial resistance (AMR) of *Salmonella* isolates from many kinds of food, particularly pork and chicken in retail, in Taiwan between January 2017 and December 2019. The E-test was used to assess antimicrobial susceptibility and a polymerase chain reaction was performed for serotyping. A total of 459 different foods were investigated, and 117 *Salmonella* strains were isolated. Retail pork and chicken were the most common *Salmonella*-contaminated foods (64.1% and 29.1%, respectively). Of the 117 isolates, 23 serotypes were identified. The serotypes Derby (16.2%), Anatum (13.7%), and Agona (8.5%) were the most prevalent. The resistance rates to ciprofloxacin, ceftriaxone, and carbapenem were 41.9%, 11.1%, and 1.7%, respectively. The Derby and Anatum serotypes were prevalent in chicken and pork; the Anatum serotype had significantly higher ciprofloxacin and ceftriaxone resistance rates and was highly prevalent in 2017 and 2018. Multi-locus sequence typing analysis revealed that the 58 randomly chosen *Salmonella* isolates belonged to 18 sequence types (STs). ST64 (Anatum, 16 out of 58, 27.6%) was the most common, followed by ST321 (Muenster, 6/58, 10.3%), ST831 (Give, 5/58, 8.6%), ST155 (London, 4/58, 6.9%) and ST314 (Kentucky, 4/58, 6.9%). Multidrug-resistant *Salmonella* strains were remarkably observed in the serotypes Anatum (ST64) and Goldcoast (ST358). This study revealed that retail pork was commonly contaminated with antimicrobial-resistant *Salmonella*. Thus, periodic investigations of *Salmonella* serotypes and AMR are needed.

## 1. Introduction

*Salmonella* is one of the most common pathogens causing foodborne illnesses in humans, such as acute gastroenteritis, bacteremia, meningitis, arthritis and mycotic aneurysm, and the World Health Organization classifies it as a significant cause of illness and death [1]; however, many animals, such as pigs, chickens, rodents, and cattle, can carry *Salmonella* in their digestive tract and even invade and multiply in enterocytes and tonsillar lymphoid tissues without significant signs of illness [2]. Non-typhoidal salmonellosis (NTS) causes 78 million foodborne infections and 59,000 deaths each year, and NTS infection has become an emerging issue in public health worldwide [1]. Eating *Salmonella*-contaminated foods, such as poultry, pork, beef, and eggs, causes most *Salmonella* illnesses in humans [3]. The burden imposed by NTS is complicated in some infections caused by multidrug-resistant (MDR) bacteria [4]. *Salmonella* has increasingly emerged, and there is growing concern about its resistance to last-line therapies [5].

A variety of foods linked to *Salmonella* outbreaks have been extensively studied [6,7]. Various resistance genes have been identified in *Salmonella* from different sources, with NTS infections posing a significant threat to global human health and being a cause of foodborne illnesses [8]. NTS infection is also the leading cause of foodborne illness in Taiwan [9]. According to Taiwanese research, antimicrobial resistance (AMR) is a serious problem in clinical *Salmonella* isolates [10]. The antimicrobial-resistant *Salmonella* spreads through the food chain, especially the MDR serotypes Anatum and Goldcoast in Taiwan [11,12], and it has been making the treatment of *Salmonella* infections in clinical practice more difficult [13].

However, there is little information on the epidemiology and AMR of *Salmonella* isolated from food in Taiwan, making it difficult to assess the impact of food-related AMR on public health [14]. Understanding the status of *Salmonella* contamination in food, as well as AMR, is critical for effectively controlling the spread of *Salmonella*. To better understand antimicrobial-resistant *Salmonella* isolated from food, we evaluated *Salmonella* contamination and AMR in food from Taiwanese markets.

## 2. Materials and Methods 

### 2.1. Sample Collection and Identification of Salmonella

We collected *Salmonella* isolates from food samples. All the samples were collected from New Taipei City and Taoyuan City in Northern Taiwan. These two densely populated districts are the main sources of patients treated at Chang Gung Memorial Hospital (CGMH). We randomly gathered food samples from 16 traditional markets and supermarkets from 2017 to 2019. The samples were processed using the methods suggested by the Association of Official Agricultural Chemists International and specified in the Microbiology Laboratory Guidebook of the United States Department of Agriculture/Food Safety and Inspection Service [15]. Each sample weighed 200–250 g and was collected at the point of sale in vendor-supplied containers, much like a consumer would purchase food. The packed pieces were immediately placed in a cooler and transferred to the laboratory, where they were stored at 4 °C until evaluation; the holding time did not exceed 16 h.

### 2.2. Microbiological Analysis

The 25 g samples (e.g., giant vegetables, fruits, or meat) were broken up in a homogenizer and incubated in 225 mL of BBL™ Gram-Negative Broth (Becton Dickinson Biosciences, Franklin Lakes, NJ, USA) for 24 h at 37 °C, followed by sub-incubation of a 0.1 mL aliquot of primary cultured bacterial solution in 10 mL fresh Rappaport-Vassiliadis medium (Becton, Dickinson and Co.) for 24 h at 42 °C. The *Salmonella* strains were obtained from the bright-black single colonies using *Salmonella*-selective HardyCHROM™ SS NoPro Agar plates (Hardy Diagnostics, Santa Maria, CA, USA). *Salmonella* serotyping was conducted using the multiplex polymerase chain reaction (mPCR) method, as described previously [16]. Bacterial genomic DNA was extracted using DNeasy Blood and Tissue Kits (Qiagen, Hilden, Germany), and an mPCR test was applied using the primer set design based on the chromosomal sequences of *S. enterica* serovars Typhimurium LT2 (STM), Typhi CT18 (STY) and Enteritidis (PT4) [16]. The serotype of each *Salmonella* isolate was determined based on its PCR amplicon pattern analyzed using electrophoresis in 2.5% agarose gel and compared to the patterns of the 30 most prevalent *S. enterica* serotypes. 

### 2.3. Antimicrobial Susceptibility Testing

We tested the *Salmonella* isolates purified from the food samples for resistance to antimicrobials. The antimicrobial sensitivity of the isolates to ciprofloxacin (susceptible: ≤0.5 μg/mL; resistant: ≥1 μg/mL), ceftriaxone (susceptible: ≤8 μg/mL; resistant: ≥64 μg/mL) and ertapenem (susceptible: ≤1 μg/mL; resistant: ≥4 μg/mL) was determined using E-test strips according to the Clinical and Laboratory Standards Institute’s (https://www.nih.org.pk/wp-content/uploads/2021/02/CLSI-2020.pdf; accessed on 14 April 2022) instructions [17].

### 2.4. Multi-Locus Sequence Typing (MLST) and Sequence Data Analyses

MLST was performed using PCR amplification and using 7 specific primer sets designed according to 7 housekeeping genes (*aroC*, *dnaN*, *hemD*, *hisD*, *purE*, *sucA*, and *thrA*), following the methods reported by Jolley et al. and referred to on the website of the *Salmonella enterica* MLST Database (https://pubmlst.org/bigsdb?db=pubmlst_mlst_seqdef&page=schemeInfo&scheme_id=2; accessed on 14 April 2022) [18]. The sequence type of each *Salmonella* isolate was determined according to the identity of 7 allelic profiles through sequencing PCR amplicons of 7 loci and screening with known sequences registered in the MLST Database. 

### 2.5. Statistical Analysis

The prevalence of AMR and replicon types were expressed as percentages of the total number of *Salmonella* isolates and calculated using Microsoft Excel (Microsoft Inc., Redmond, WA, USA). A chi-square test of independence was used to examine the correlations between different years of *Salmonella* collection for resistance phenotypes using SPSS version 22.0 software (SPSS Inc., Chicago, IL, USA). A *p*-value < 0.05 was considered significant.

## 3. Results

Of the 459 food samples analyzed, 117 (25.4%) were positive for *Salmonella*. The foods investigated included pork (31.8%), vegetables (20.9%), chicken (20.3%), eggs (10 %), seafood (3.2%), fruit (5.0%), beef (5.8%), delicatessen (2.1%), sauces (0.4%), pig intestines (0.4%), and duck (0.2%). *Salmonella* was significantly more prevalent (*p* < 0.001) in pork (51.3% of the samples were positive) and chicken (35.2%) than in seafood (6.6%), vegetables (5.2%), beef (3.7%), eggs (0%), or other foods (2.7%). Twenty-three serotypes were identified from the 117 *Salmonella*-positive samples (Table 1). The Derby serotype was predominant (16.2%), followed by the Anatum (13.7%), Agona (8.5%), Albany (7.7%), Kentucky (7.7%), and London (6.8%) serotypes; these serotypes accounted for 60.6% of the isolates (Table 1 and Table 2). The prevalence of other serotypes ranged from 0.9% to 5.1%. Pork purchased from the supermarket had less *Salmonella* contamination than that purchased from a traditional market. The five serotypes most often identified among the pork isolates were Derby, Anatum, Agona, London, and Give. The Albany, Kentucky (12.8%), Anatum, and Brancaster (5.1%) serotypes were the most commonly isolated serotypes from chicken. In addition, the *Salmonella* serotypes Agona, Corvallis, Give Goldcoast, Livingstone, London, Mbandaka, Newport, Potsdam, Rissen, and Weltevreden were found in pork only, and other serotypes, including Brancaster, Enteritidis, Schwarzengrund, and Thompson, were isolated from chicken only. *Salmonella* Kaitaan and *Salmonella* Zigong were only detected in vegetable samples. The Derby serotype was the most common among the pork isolates (*n* = 16), while the Kentucky (*n* = 6) and Albany (*n* = 6) serotypes were isolated from chicken. The Anatum serotype was most prevalent in pork, followed by chicken and beef. 

The AMR rate from 2017 to 2019 is shown in Figure 1. The resistance rates of ciprofloxacin and ceftriaxone were higher in 2017 (60.7% and 17.9%, respectively), particularly the ciprofloxacin-resistant rate between 2017 and 2018, with a significant difference (*p* = 0.027). Table 1 and Table 2 list the resistance rates of the 117 *Salmonella* isolates. The resistance rates to ciprofloxacin, ceftriaxone, and carbapenem were 41.9%, 11.1%, and 1.7% in total, respectively (Table 2). Of all the *Salmonella* isolates tested, 50 (42.7%) were resistant to at least one highly antimicrobial-resistant agent and 13 (11.1%) were resistant to both (Table 1). Resistance to carbapenem was observed in two isolates. The carbapenem-resistant isolates included the Anatum serotype from pork and the Derby serotype from chicken. The resistance rate differed depending on the serotype. The Goldcoast and Anatum serotypes had the highest resistance to ceftriaxone (100% and 68.7%, respectively). The resistance rates to ciprofloxacin were higher among the Brancaster (100%), Give (100%), Goldcoast (100%), Albany (77.8%), Anatum (75.0%), London (50%), Kentucky (44.4%), Derby (42.1%), Enteritidis (25.0%), and Livingstone (20.0%) serotypes, while the Agona, Muenster, Newport, Typhimurium, and Weltevreden serotypes had lower resistance rates to ciprofloxacin and ceftriaxone. Of the 117 *Salmonella* isolates, 58 were randomly chosen for MLST analysis (Table 1). Eighteen STs were identified. The most common ST was ST64 (serotype Anatum, 16/58, 27.6%; represented by 16 *Salmonella* isolates among 58 MLST-confirmed isolates), followed by ST321 (Muenster, 6/58, 10.3%), ST831 (Give, 5/58, 8.6%), ST155 (London, 4/58, 6.9%), ST314 (Kentucky, 4/58, 6.9%), ST358 (Goldcoast, 2/58, 3.4%), ST31 (Newport, 2/58, 3.4%), and ST96 (Schwarzengrund, 2/58, 3.4%). Most of the STs obtained in this study were correlated with particular serotypes, such as ST64 with Anatum, ST831 with Give, and ST358 with Goldcoast; moreover, Kentucky was correlated to at least two sequence types, ST198 and ST314 (Table 1).

## 4. Discussion

Our study provides the status of *Salmonella* contamination in food and the AMR properties in Taiwan. We found a high prevalence of *Salmonella* in retail pork. High AMR was found in at least one-third of the *Salmonella* isolates from chicken and pork, and was related to specific serotypes. Furthermore, we noted the diversity in the serotypes and genotypes of the *Salmonella* isolates during the 3 years.

In the current investigation, the prevalence rates of *Salmonella* in pork and chicken samples were 51.3% and 35.2%, respectively, which were consistent with previous studies conducted in Taiwan [9,19,20]. The *Salmonella* strains obtained from pork carcasses in Taiwan were previously relatively resistant to the antimicrobial treatments tested [21,22,23]. This study found *Salmonella* in 51.3% of retail pork. This figure is higher than the 2.1–31% reported in Ireland, Denmark, China, and the United States [24,25,26,27,28]. The relatively high contamination rate of *Salmonella* may be occurring because retail pork is generally obtained from traditional markets without consistent refrigerated storage. Additionally, Taiwan is in a subtropical zone, and *Salmonella* thrives in hot weather. In the current study, Derby was the most prevalent *S. enterica* serotype (abbreviated as *S.* Derby), matching previous results from other regions and nations [29,30]. *S.* Derby is one of the top ten serotypes isolated from human salmonellosis infections in many countries, including Taiwan [14,30,31,32]. In addition, a large *S.* Derby clone spread from pigs to humans in Europe [33,34]. According to these investigations, *S.* Derby has the potential to infect people through pork products, indicating that *Salmonella* may be transmitted from pigs to humans via the food chain.

The *Salmonella* strains isolated from pork in this study were relatively resistant to the antimicrobial agents tested, compared to the results from pork carcasses studied in Taiwan between 2000 and 2003, all of which were susceptible to ceftriaxone. Ceftriaxone-resistant *Salmonella* isolates from humans have emerged in Taiwan. In this study, high AMR was common in some *Salmonella* serotypes detected in pork samples, particularly the Anatum and Goldcoast serotypes. A human outbreak of MDR *S.* Anatum emerged in Taiwan in 2015 [11]. This serotype is virulent in humans, and children became ill more frequently due to contamination from meat. Pork and poultry have been identified as potential carriers. This serotype is responsible for foodborne diseases, and it has caused outbreaks in many countries. *S.* Goldcoast infection was first reported in Taiwan in 2014, and all but one isolate were pan-susceptible from 2014 to 2016. *S.* Goldcoast infections spiked in 2018, and all isolates were MDR. According to whole-genome sequencing, the clone responsible for the clinical outbreak shared its genealogy with an *S*. Goldcoast strain detected in a retail meat isolate [12]. The antimicrobial-resistant isolates of serotypes Anatum and Goldcoast from Taiwan, which harbor conjugatable plasmids (such as pSal-3973_DHA_CMY and pSal-5364) with *qnrB4* (or *qnrS1*) and *bla*_DHA-1_ (or *bla*_CMY-2_) genes conferring resistance to ciprofloxacin and ceftriaxone, respectively, show high genetic similarity to Anatum isolates from the UK and Goldcoast isolates from the UK (strain 296839) and Germany (strain SAL_IB6386AA) [11,12]. Notably, very few MDR isolates were verified before 2015 [11,12,35,36]. Such global-wide dissemination of antimicrobial-resistant *Salmonella* strains via bacterial conjugation to transmit the MDR gene-carrying plasmids could be the reason why certain areas have isolates that are resistant to certain antimicrobial agents.

Overall, *Salmonella* was detected in 35.2% of chicken samples. The prevalence was higher than in a previous report (25.1%; 95% confidence interval: 21.7–28.7). Similar to previous studies [37], Albany and Kentucky are the most common serotypes in chicken. *S.* Albany dominates chicken abattoirs in Taiwan. The Kentucky serotype is the most common in poultry products from North America [38]. The Agona serotype is susceptible to all antimicrobial agents. The Kentucky serotype is also sensitive to ceftriaxone. The link between serotypes and human epidemics in the United States is very weak, with rates < 1% [39]. 

Our results revealed that ST64 among the 18 identified STs was the most frequent *Salmonella* genotype in the current study. ST64 was commonly detected in Asian and European patients, and occurs widely in pork from Europe and the USA [40]. ST34 and ST19 were the second and third most common serotypes, respectively, widely found in humans and pigs in Japan, China, the USA, and Europe, indicating that *Salmonella* may be transmitted from pigs to humans via the food chain [24,41]. Additionally, ciprofloxacin-resistant ST314 strains have been increasing since 2014 (from 5.6% in 2014 to 53.2% in 2016). The emergence of ST314 strains increasingly becoming ciprofloxacin-resistant, similar to ST198, is a warning in global health. The recent emergence of concurrent resistance to ciprofloxacin and ceftriaxone among *Salmonella* isolates is posing a serious threat that requires further monitoring. Another emerging issue related to colistin resistance that was first reported in patients with infection of the *Salmonella* ST155 strain harboring a colistin-resistant *mcr-1* gene in China highlights the importance of cautious use and continuous monitoring of colistin resistance in both clinical and veterinary medicines [41,42].

Our findings reveal a high prevalence of *Salmonella* in retail meat in Northern Taiwan. Strong AMR was found in half of the *Salmonella* isolates from chicken and one-third of the *Salmonella* isolates from pork. High AMR correlates to specific serotypes, particularly MDR Anatum and Goldcoast, posing a danger to the control of *Salmonella* infections in humans and animals. More epidemiological surveillance is needed in Taiwan to determine the prevalence, AMR, and subtyping characteristics of foodborne pathogens, which will help to develop scientifically sound public health policies and put into place practical actions to ensure the safety of our food supply.

## Figures and Tables

**Figure 1 pathogens-11-00705-f001:**
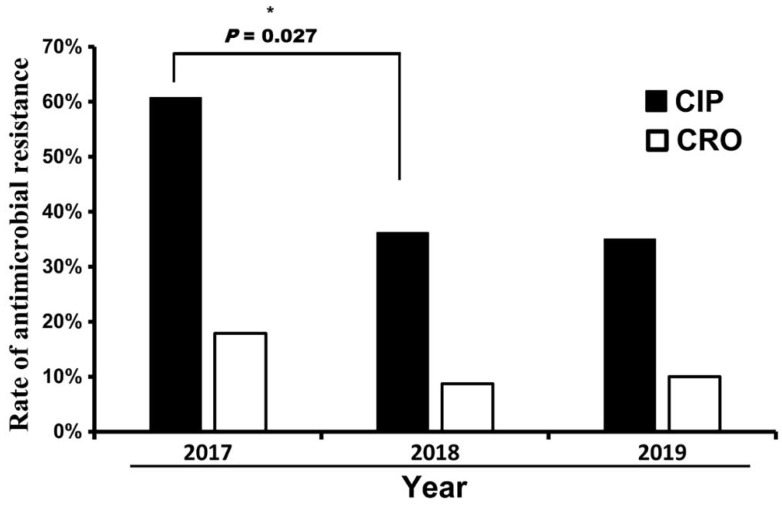
Change in the antimicrobial resistance rates among *Salmonella* isolates from 2017 to 2019. CIP, ciprofloxacin; CRO, ceftriaxone. *: significant difference with *p* < 0.05.

**Table 1 pathogens-11-00705-t001:** Diversity profiles of the *Salmonella* isolates based on MLST, serotyping, and antimicrobial resistance.

Serotypes	Sequence Type *	No.	Antimicrobial Resistance	No.	Resistance Rate (%)	Source (No.)
Agona	ND	10	S	10	0	Pork (10)
Albany	ST292	1	S	1	77.8	Chicken (1)
	ND	8	S	1		Chicken (1)
			CIP	7		Chicken (4); Vegetables (1); Giblets (1); Clams (1)
Anatum	ST64	16	S	4	75.0	Pork (4)
			CIP, CRO	11		Pork (4); Chicken (4); Haslets (2); Beef (1)
			CIP, CRO, ETP	1		Haslets (1)
Brancaster	ST2133	1	CIP	1	100	Chicken (1)
	ND	2	CIP	2		Chicken (2)
Corvallis	ST1541	1	S	1		Pork (1)
Derby	ND	19	S	10	47.4	Pork (8); Vegetables (2)
			CIP	8		Pork (6); Haslets (2)
			ETP	1		Chicken (1)
Enteritidis	ND	4	S	3	25.0	Chicken (3)
			CIP	1		Chicken (1)
Give	ST831	5	CIP	5	100	Pork (4); Haslets (1)
Goldcoast	ST358	2	CIP, CRO	2	100	Pork (1); Chicken (1)
Kaitaan	*	1	S	1	0	Vegetables (1)
Kentucky	ST198	1	S	1	44.4	Chicken (1)
	ST314	4	S	1		Pork (1)
			CIP	3		Chicken (3)
	ND	4	S	3		Chicken (2); Pork (1)
			CIP	1		Pork (1)
Livingstone	ND	5	S	4	20.0	Chicken (2); Pork (2)
			CIP	1		Pork (1)
London	ST155	4	CIP	4	50.0	Pork (4)
	ND	4	S	4		Pork (4)
Mbandaka	ND	2	S	2	0	Pork (2)
Muenster	ST321	6	S	6	0	Pork (3); Chicken (2); Haslets (1)
Newport	ST31	2	S	2	0	Pork (2)
Potsdam	ST2462	1	S	1	0	Pork (1)
Rissen	ST469	1	S	1	0	Pork (1)
Schwarzengrund	ST96	2	CIP	2	100	Chicken (2)
Thompson	ST26	1	S	1	0	Chicken (1)
Typhimurium	ST36	1	S	1	0	Chicken (1)
	ND	4	S	4		Pork (3); Chicken (1)
Weltevreden	ST65	1	S	1	0	Pork (1)
	ND	3	S	3		Pork (3)
Zigong	ST3467	1	S	1	0	Vegetables (1)
Total		117		117	42.7%	117

No., number; ND, not detected; CIP, ciprofloxacin; CRO, ceftriaxone; ERT = ertapenem; * no MLST analysis but PFGE typing to confirm that it was *S*. Kaitaan.

**Table 2 pathogens-11-00705-t002:** Antimicrobial resistance profiles of the *Salmonella* serotypes.

Serotypes *	No. of Isolates (%)	Ciprofloxacin Resistance Rate %	Ceftriaxone Resistance Rate %	Carbapenem Resistance Rate %
Derby	19 (16.2%)	42.1% (8/19)	0% (0/19)	5.3% (1/19)
Anatum	16 (13.7%)	75% (12/16)	68.8% (11/16)	6.3% (1/16)
Agona	10 (8.5%)	0% (0/10)	0% (0/10)	0% (0/10)
Kentucky	9 (7.7%)	44.4% (4/9)	0% (0/9)	0% (0/9)
Albany	9 (7.7%)	77.8% (7/9)	0% (0/9)	0% (0/9)
London	8 (6.8%)	50% (4/8)	0% (0/8)	0% (0/8)
Muenster	6 (5.1%)	0% (0/6)	0% (0/6)	0% (0/6)
Give	5 (4.3%)	100% (5/5)	0% (0/5)	0% (0/5)
Livingstone	5 (4.3%)	20% (1/5)	0% (0/5)	0% (0/5)
Typhimurium	5 (4.3%)	0% (0/5)	0% (0/5)	0% (0/5)
Enteritidis	4 (3.4%)	25% (1/4)	0% (0/4)	0% (0/4)
Weltevreden	4 (3.4%)	0% (0/4)	0% (0/4)	0% (0/4)
Brancaster	3 (2.6%)	100% (3/3)	0% (0/3)	0% (0/3)
Goldcoast	2 (1.7%)	100% (2/2)	100% (2/2)	0% (0/2)
Mbandaka	2 (1.7%)	0% (0/2)	0% (0/2)	0% (0/2)
Newport	2 (1.7%)	0% (0/2)	0% (0/2)	0% (0/2)
Schwarzengrund	2 (1.7%)	100% (2/2)	0% (0/2)	0% (0/2)
Corvallis	1 (0.9%)	0% (0/1)	0% (0/1)	0% (0/1)
Kaitaan	1 (0.9%)	0% (0/1)	0% (0/1)	0% (0/1)
Potsdam	1 (0.9%)	0% (0/1)	0% (0/1)	0% (0/1)
Rissen	1 (0.9%)	0% (0/1)	0% (0/1)	0% (0/1)
Thompson	1 (0.9%)	0% (0/1)	0% (0/1)	0% (0/1)
Zigong	1 (0.9%)	0% (0/1)	0% (0/1)	0% (0/1)
Total	117	41.9% (49/117)	11.1% (13/117)	1.7% (2/117)

* The order of serotype is exhibited based on its number of isolates.

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
