# Peer review of "Prevalence, Serotypes, and Antimicrobial Resistance Patterns of Non-Typhoid Salmonella in Food in Northern Taiwan"

_pathogens, 2022, doi:10.3390/pathogens11060705_

Round 1

Reviewer 1 Report

some suggests :

1, in keywords , why was  only the  retail pork listed?  cheken why not ?

2, many Salmonella serotypes had been identified and  listed,  MLST shall be used to  differentiate same Salmonella serotype, 

3,  why was Non-typhoid Salmonella highlighted ?    

4, food in the topic was highlighted,  it shall say that  many kind of food , but only two kinds , chicken and pork in retail

5, Multi-drug resistance shall be summarized, because it was focsed on by every one 

6,line 31, "Salmonella is one of the most common foodborne illnesses"     Salmonella is not a illness,  the logic  is not correct.  I fell thet "Salmonella is one of the most common foodborne panthogens"  and so, please check around the paper .

7,  Table 1, why was the number  2/97 , I feel that it shall be 2/117?  because the total number are 117

8, all antibotics in Antimicrobial Susceptibility Testing were listed. only two kinds?

9, Figure 1,  in Y-axis  "resiatane" is "resiatance" ?

10, please check this manscript the logic and expression  for better show 

Author Response

Response to the reviewers’ comments

Reviewer 1

Comments and Suggestions for Authors

some suggests :

1, in keywords , why was  only the  retail pork listed?  chicken why not ?

Reply: In Keywords, we have added chicken, following pork. In addition, we have amended the terms of “Pork and poultry” to “Retail pork and chicken” for clarity.

2, many Salmonella serotypes had been identified and listed.  MLST shall be used to differentiate same Salmonella serotype, 

Reply: Serotypes and antimicrobial resistance patterns among all Salmonella (117 isolates) were identified in this study, but MLST analysis was not carried out for all isolates. Only 58 isolates were randomly chosen for MLST. The description has been rephrased on lines 153‒161, pages 9‒10.

3,  why was Non-typhoid Salmonella highlighted ?    

Reply: We have added a description to highlight the importance of NTS on line 47, page 3.

4, food in the topic was highlighted, it shall say that many kinds of food, but only two kinds , chicken and pork in retail

Reply: We have amended the description of “…from food in Taiwan…” into “…from many kinds of food, particularly pork and chicken in retail, in Taiwan…” for clarity on line 21, page 2.

5, multi-drug resistance shall be summarized, because it was focused on by every one 

Reply: We have summarized more descriptions about multi-drug-resistance on lines 34, 35 and 186‒203.

6, line 31, "Salmonella is one of the most common foodborne illnesses" Salmonella is not an illness, the logic is not correct. I fell that "Salmonella is one of the most common foodborne pathogens" and so, please check around the paper .

Reply: Thanks for comments. We have rephrased the description into “Salmonella is one of the most common pathogens causing foodborne illnesses” and added more description on lines 41‒47, page 4.

7,  Table 1, why was the number  2/97 , I feel that it shall be 2/117?  because the total number are 117

Reply: The total number in Table 1 has been corrected to 117. In addition, serotypes, number and ratio (%) have been amended as well.

8, all antibotics in Antimicrobial Susceptibility Testing were listed. only two kinds?

Reply: The third antimicrobial susceptibility test using ertapenem (susceptible: £ 1 mg/mL; resistant: ≥ 4 mg/mL) has been added in the revised manuscript on lines 98-99, page 6.

9, Figure 1,  in Y-axis  "restane" is "resistance" ?

Reply: In Figure 1, the subtitle of Y-axis has been corrected from "restane" to "resistance"

10, please check this manuscript the logic and expression  for better show 

Reply: English editing has been done in this revised manuscript, and many descriptions have been rephrased for clarity and better expression according to reviewers’ comments. All changes have been highlighted in red.

Reviewer 2 Report

  1. Line 43: This antimicrobial-resistant 43
    Salmonella could not only spread to people through the food chain, but it could also make 44
    treating Salmonella infections in clinical practice more difficult 
    Rephrase since a particular strain of Salmonella is not being pointed out here.
  2. The discussion needs to be improvised. Author's thoughts with evidence on why certain areas have isolates resistant to certain antibiotics is imperative. Such studies are well substantiated with  Principal Component Analysis to  pin or propose certain underlying reasons why the abundance is seen in such a way in a particular area.
    Antibiotic use rather misuse has been a huge concern that has resulted in certain areas having more resistance than others. 
    Discuss the results with this aspect.

Author Response

Response to the reviewers’ comments

Reviewer 2

  1. Line 43: This antimicrobial-resistant 43
    Salmonella could not only spread to people through the food chain, but it could also make 44-treating Salmonella infections in clinical practice more difficult 
    Rephrase since a particular strain of Salmonella is not being pointed out here.

Reply: Thanks for comments. We have added particular MDR strains of Salmonella on lines 57‒59 of page 4, where the sentence has been changed to “The antimicrobial-resistant Salmonella spreads through the food chain, especially MDR serotypes Anatum and Goldcoast in Taiwan [11,12] and it has been making the treatment of Salmonella infections in clinical practice more difficult [13]”.

  1. The discussion needs to be improvised. Author's thoughts with evidence on why certain areas have isolates resistant to certain antibiotics is imperative. Such studies are well substantiated with Principal Component Analysis to pin or propose certain underlying reasons why the abundance is seen in such a way in a particular area.
    Antibiotic use rather misuse has been a huge concern that has resulted in certain areas having more resistance than others. 
    Discuss the results with this aspect.

Reply: Thanks for comments. We have added more description in section of Discussion about why certain areas have isolates resistant to certain antibiotics on lines 186‒203, page 11. In addition, more description on Salmonella isolates with different sequence types have been added on lines 217‒223, pages 12‒13.

Reviewer 3 Report

This study aimed to evaluate Salmonella contamination and AMR in food from Taiwanese markets

  1. The introduction is short, you need to give more emphasis on AMR in Salmonella, clinical signs in humans and animals, the importance of the poultry and pork industry in Taiwan, and so on.
  2. In section 2.1: add more details about the collected samples such as: what kind of samples and their distribution, number, season other available data. 
  3. add a table in section 2.1 containing your sample distribution.
  4. What kind of Gram-Negative Broth was used?
  5. Provide more details about PCR methods used for Salmonella confirmation, primers, and conditions.
  6. Provide more details on MLST and Sequence Data Analysis.
  7. provide the results of the prevalence in a bar graph and run stats for prevalence difference.
  8. I suggest moving a serotyping table before the AMR table.
  9. It would be more interesting if you provide the genotypic resistance data and compare it with the phenotypic resistance.

Author Response

Response to the reviewers’ comments

Reviewer 3

This study aimed to evaluate Salmonella contamination and AMR in food from Taiwanese markets

  1. The introduction is short, you need to give more emphasis on AMR in Salmonella, clinical signs in humans and animals, the importance of the poultry and pork industry in Taiwan, and so on.

Reply: Thanks for comments. We have added more descriptions on AMR in Salmonella, clinical signs in humans and animals, and the importance of the poultry and pork industry, on lines 41‒59, page 4.

  1. In section 2.1: add more details about the collected samples such as: what kind of samples and their distribution, number, season other available data. 

Reply: We have briefly added the Salmonella-positive sources in Table 1.

  1. add a table in section 2.1 containing your sample distribution.

Reply: We have simply added the information of Salmonella-positive sources in Table 1, but the sample distribution is not inclued because the collected foods purchased from many stores during 2017‒2019 were very complicated.

  1. What kind of Gram-Negative Broth was used?

Reply: We have amended the term into “BBL™ Gram-Negative Broth (Becton Dickinson Biosciences, Franklin Lakes, NJ, USA)” on lines 82-83, page 6.

  1. Provide more details about PCR methods used for Salmonella confirmation, primers, and conditions.

Reply: We have amended the description of mPCR methods for Salmonella confirmation on lines 87‒94, page 6.

  1. Provide more details on MLST and Sequence Data Analysis.

Reply: We have amended the description of “MLST and Sequence Data Analysis” in detail on lines 103‒110, page 7.

  1. provide the results of the prevalence in a bar graph and run stats for prevalence difference.

Reply: The prevalence difference in our bar graph (Figure 1) has been indicated and described on lines 139-141, pages 8‒9.

  1. I suggest moving a serotyping table before the AMR table.

Reply: We have changed the order of Tables 1 and 2 on pages 23 and 24 as the reviewer’s suggestion.

  1. It would be more interesting if you provide the genotypic resistance data and compare it with the phenotypic resistance.

Reply: We did not conduct genotypic resistance data for all Salmonella isolates but only the data for serotypes Anatum and Goldcoast had been completed and published, as shown in References 12 and 13.

Round 2

Reviewer 3 Report

The authors addressed all the comments, however, they still need to provide the sample source or the kind of collected samples

Author Response

Reviewer 3: The authors addressed all the comments, however, they still need to provide the sample source or the kind of collected samples

Response:

Many thanks to reviewer 3 for the valuable comments.

1.For collection source, as shown on lines 64‒66.
2.For names of collected samples, as shown on lines 115‒120.
3.For names and number of collected samples, as shown on the most
left-side column of Table 1, line 137.
